# Analysis of Changes in Antibiotic Use Patterns in Korean Hospitals during the COVID-19 Pandemic

**DOI:** 10.3390/antibiotics12020198

**Published:** 2023-01-18

**Authors:** Bongyoung Kim, Hyeonjun Hwang, Jungmi Chae, Yun Seop Kim, Dong-Sook Kim

**Affiliations:** 1Department of Internal Medicine, Hanyang University College of Medicine, Seoul 04763, Republic of Korea; 2Graduate School of Data Science, Kyungpook National University, Daegu 41566, Republic of Korea; 3Health Insurance Review & Assessment Service, Wonju 26465, Republic of Korea; 4The Office of Research Strategy, Korea University Medical Center, Seoul 06014, Republic of Korea

**Keywords:** COVID-19, antibiotics, antibiotic use, Korea

## Abstract

With the onset of the coronavirus disease 2019 (COVID-19) pandemic, changes in patient care and antibiotic use have occurred in hospitals. The data of the National Health Insurance System’s claims of inpatients from all hospitals in Korea between January 2019 and December 2020 were obtained from the Health Insurance Review & Assessment Service and analyzed. The trend in the use of all antibacterial agents in both hospitals declined for the total number of COVID-19 patients at the bottom 10% and those in the top 10%. Specifically, a decreasing trend in the use of broad-spectrum antibacterial agents predominantly prescribed for community-acquired cases and narrow-spectrum beta-lactam agents were observed in both hospitals. In the aftermath of the COVID-19 pandemic, the total use of antibacterial agents has gradually decreased among patients with pneumonia and those with severe COVID-19. In contrast, its use has increased gradually among those with mild to moderate COVID-19. A decreasing trend in overall antibiotic use was observed during the COVID-19 pandemic, and an increasing trend in antibiotic use was observed in patients with mild to moderate COVID-19 in Korean hospitals.

## 1. Introduction

In Korean hospitals, overall antibiotic use, especially that of broad-spectrum antibiotics, has increased over the past decade [1,2,3]. The most plausible explanation for this phenomenon may be attributed to the increase in the proportion of cases caused by antibiotic-resistant pathogens [4,5]. To minimize unnecessary antibiotic use and curb the potential threat of antimicrobial resistance (AMR), the Korean government has launched the Korean National Action Plan on Antimicrobial Resistance 2016–2020, and since then, each hospital has been making efforts to implement antimicrobial stewardship programs (ASPs) [6].

However, with the onset of the coronavirus disease 2019 (COVID-19) pandemic, changes in patient care and antibiotic use have occurred in hospitals. The reduced utilization of healthcare services due to the shortage of medical resources, fear of transmission of COVID-19, and the reduction in overall community-acquired communicable cases due to social distancing could have led to the decrease in antibiotic use in hospitals [7,8]. However, minimization or suspension of other quality improvement programs, including ASP and infection control programs against AMR, as well as focusing the capabilities of the hospitals on responding to the COVID-19 pandemic, may offset the reduction in antibiotic use [9,10]. Furthermore, a surge in COVID-19 cases which could have similar clinical features to bacterial pneumonia might influence the antibiotic usage pattern [11]. For instance, when physicians face critically ill COVID-19 patients, and the diagnosis of bacterial superinfections is uncertain, some of them might make a decision to use broad-spectrum antibiotics immediately rather than wait for a while [12].

The objective of this study is to analyze the changing patterns in antibiotic use in Korean hospitals during the COVID-19 pandemic.

## 2. Materials and Methods

### 2.1. Data Source

The National Health Insurance System of Korea covers approximately 98% of the total population, including low-income families in Korea [13]. We obtained the National Health Insurance System’s claim data for inpatients admitted to all primary, secondary, tertiary, and long-term care hospitals in Korea between January 2019 and December 2020 from the Health Insurance Review & Assessment Service (HIRA). The data included information on age, sex, insurance type, clinic or hospital code, the area code of the clinic or hospital, care type (inpatient or outpatient), start date of treatment, treatment duration, admission or visit days, primary discharge diagnosis code, sub-discharge diagnosis code, medical department in charge, medical cost, medical or surgical treatments, and prescribed pharmaceuticals. The discharge diagnoses were coded according to the *International Classification of Diseases, Tenth Revision (ICD-10).*

### 2.2. Definitions

We defined antibiotic medication according to the Anatomical Therapeutic Chemical class J01, and antifungal, antiviral, or antiparasitic drugs were not included [14]. Of the antibiotics, we included agents with oral or parenteral administration routes while excluding topical agents. The defined daily dose (DDD), following the ATC classification system of the World Health Organization, was adopted as a measurement of antibiotic consumption and was standardized per 1000 patient days. We adopted the antibiotic classification used for the Korea National Antimicrobial Use Analysis System and added some categories after discussion with the research group (Appendix A) [15].

‘COVID-19 patient management hospitals’ were defined as those hospitals that managed more than 10 COVID-19 patients during the period. Hospitals that managed 10–20 COVID-19 patients in total were classified as the bottom 10% (BOTTOM 10% hospitals), whereas hospitals that managed 954 or more COVID-19 patients were in the top 10% (TOP 10% hospitals) (Appendix A). Patients with ICD-10 codes relevant to COVID-19 in the first- or second-priority discharge diagnoses were defined as COVID-19 patients. Those with a history of high-flow nasal cannula oxygen therapy, ventilator use, continuous renal replacement therapy, or extracorporeal membrane oxygenation were classified as ‘severe COVID-19 patients’, and those who had no such history were the ‘mild-to-moderate COVID-19 patients’. Patients with pneumonia were defined according to the ICD-10 codes for all-cause pneumonia in the first- or second-priority discharge diagnoses (Appendix A) [16]. Given that empirical antibiotics are likely to be used for any type of pneumonia before the causative pathogens are revealed, we also included the codes relevant to viral pneumonia.

### 2.3. Statistical Analysis

We used segmented regression analysis of an interrupted time series with adjustment for autocorrelation to analyze the changing patterns of antibiotic use in hospitals [17]. The ‘change in level’ was defined as the difference between the observed value at the beginning of each period. This implied that the total effects were explained by decomposition into two changes. We defined the ‘change in trend’ as the difference between the rates of change in each period. As the first case of COVID-19 was reported in Korea in January 2020, the study period was divided into ‘pre-COVID’ (from January 2019 to December 2019) and ‘post-COVID’ (from January 2020 to December 2020) phases [18].

Linear regression models were used to assess the periodic trends in antibiotic consumption among patients with COVID-19 and pneumonia during the post-COVID phase.

All statistical analyses were performed using the Newey command (considering Newey–West standard errors) in STATA version 15 (Stata Corp. LLC, College Station, TX, USA) software. Statistical significance was set at *p* < 0.05.

### 2.4. Ethics Statement

The study design was approved by the Institutional Review Board of HIRA (IRB number: 2021-016-001). The requirement for written informed consent was waived off.

## 3. Results

Table 1 shows the changing patterns of antibiotic usage in hospitals during the COVID-19 pandemic. As shown, the proportion of total antibacterial agents did not change immediately after the beginning of the COVID-19 pandemic in Korea. However, a decreasing trend over time (−9.95 DDD/1000 patient-days/month, *p* = 0.006) was observed eventually. A similar change in the pattern was observed for the use of total antibacterial agents in both the BOTTOM 10% (−8.65 DDD/1000 patient-days/month, *p* = 0.012) and TOP 10% hospitals (−15.89 DDD/1000 patient-days/month, *p* = 0.008) (Appendix A).

Immediately after the beginning of the COVID-19 pandemic, the amount of broad-spectrum antibacterial agents predominantly used for hospital-onset cases, antibacterial agents predominantly used for resistant gram-positive bacterial cases, antibacterial agents predominantly used for extensive antibiotic-resistant gram-negative bacterial cases, and carbapenem increased significantly both in the BOTTOM 10% and TOP 10% hospitals. In comparison, decreasing trends were observed for broad-spectrum antibacterial agents predominantly used for community-acquired cases and narrow-spectrum beta-lactam agents both in the TOP 10% and BOTTOM 10% hospitals (Figure 1 and Figure 2, Appendix A).

Table 2 shows the changing patterns of antibiotic usage in patients with pneumonia or COVID-19 cases during the COVID-19 pandemic. The total use of antibacterial agents decreased gradually in those with pneumonia (coefficient −9.73, *p* < 0.001) and severe COVID-19 (coefficient −34.62, *p* = 0.022). In contrast, the use gradually increased among patients with mild-to-moderate COVID-19 (coefficient 25.55, *p* = 0.043) (Appendix A).

In patients with pneumonia, the trends for broad-spectrum antibacterial agents predominantly used for community-acquired cases (coefficient −8.05, *p* = 0.012) and fluoroquinolone (coefficient −4.57, *p* = 0.005) decreased, while the use of broad-spectrum antibacterial agents predominantly prescribed for hospital-onset cases increased significantly (coefficient 4.95, *p* = 0.010) (Figure 3). As for patients with severe COVID-19, the use of carbapenem (coefficient −12.91, *p* < 0.001) and metronidazole (coefficient −1.92, *p* = 0.030) decreased (Figure 4). An increasing trend in the use of broad-spectrum antibacterial agents predominantly prescribed for hospital-onset cases was observed in patients with mild-to-moderate COVID-19 (coefficient 3.60, *p* = 0.003) (Figure 5).

## 4. Discussion

Apart from the direct impact on hospitals, due to an increase in hospitalization rates because of COVID-19 cases and transmission within facilities, the COVID-19 pandemic has affected other health conditions at the population level, including the change in the use of medications [7,8,19]. Several previous studies have highlighted the increase in antibiotic use during the COVID-19 pandemic in hospitals. A single-center study from Spain found that the use of amoxicillin/clavulanate increased during the early phase of the pandemic, followed by an increase in the use of broad-spectrum antibiotics [20]. Another single-center study from the US showed that antibiotic consumption increased even in the hospitals away from the COVID-19 epicenter during the early stages of the COVID-19 pandemic [21]. Furthermore, a significant increase in antibiotic utilization between January 2020 and May 2020, compared with the corresponding months in the previous years, was observed in the veterans hospitals in the US [22]. In comparison, significant decreases in overall antibiotic use and broad-spectrum antibiotics have been observed in Korea [23]. Our findings are in line with those of a previous study in Korea. A similar pattern was observed when the analysis was limited to ‘COVID-19 patient management hospitals’.

Successful non-pharmaceutical interventions in the early phases may provide a possible explanation for the decreasing antibiotic use patterns in Korean hospitals during the COVID-19 pandemic. Non-pharmaceutical interventions have been implemented in Korea, including strict social distancing measures since the beginning of the COVID-19 pandemic [24]. Strict social distancing norms may have contributed to the reduction in the use of medical services, resulting in fewer antibiotic prescriptions. However, the incidence of several respiratory infections, which usually prompt an antibiotic prescription, was reduced after strict social distancing, which seems to be a more plausible explanation for this phenomenon [25]. In Korea, the incidences of several respiratory infections, including chickenpox, mumps, influenza, and other viral respiratory infections, reduced significantly during the COVID-19 pandemic [8,26]. Indeed, a previous study in Korea showed that the amount of antibiotic use changes with the degree of social distancing measures [23]. Another possible explanation can be that the ASPs in Korean hospitals are not as strict compared with the hospitals in other developed countries. Unfortunately, even ASPs in large Korean hospitals are still highly dependent on one or two specialists of infectious diseases and restrictive antimicrobial programs [27]. Therefore, the COVID-19 pandemic may have not affected the ASPs to a great extent in other countries.

An increasing trend in the use of broad-spectrum antibiotics in patients with mild-to-moderate COVID-19 is another important finding of the present study. Even though the incidence of bacterial co-infection was less than 8% in patients with COVID-19, and the US National Institutes of Health guidelines do not recommend a routine antibiotic prescription for these patients, the use of broad-spectrum antibiotics, including those for anti-methicillin-resistant *S. aureus*, and anti-pseudomonal antibiotics is prevalent in Korean hospitals [9,28]. Given that the overuse of antibiotics may lead to collateral damage, including an increase in AMR, adverse effects, and medical costs, ASP should be reinforced for patients with COVID-19 [29].

This study has some limitations. First, the accuracy of the diagnosis was not sufficiently verified. As the data for National Health Insurance System claims do not include information on laboratory or radiological results, only the discharge diagnosis codes were reviewed. There is a possibility that some patients may have had other infectious diseases that required the use of antibiotics, such as catheter-associated bloodstream infections. Furthermore, some cases of pneumonia or COVID-19 may have been coded under other ICD-10, including sepsis or adult respiratory distress syndrome, and these may not have been included in our analysis. In addition, some cases of pneumonia could have been secondary complications of COVID-19 and might have been included in both the COVID-19 and pneumonia groups. Second, factors that could affect antibiotic use patterns in hospitals were not adjusted. Not only the burden of COVID-19 but also the distribution of diseases, patient severity, and hospital size were not included in the analysis, which could affect the antibiotic use pattern in the hospital. Third, the antibiotic prescription was evaluated using DDD and not according to the days of therapy. There may be a possibility of underestimation of antibiotic use among patients with kidney dysfunction or pediatric cases [30]. Finally, the information on duration or combined antibiotic therapy was unavailable due to the nature of the National Health Insurance System’s claims data. This point also prevents understanding of the timing of the initiation of antibiotics in COVID-19. A more detailed analysis of antibiotic use is warranted.

## 5. Conclusions

A decreasing trend in overall antibiotic use was observed during the COVID-19 pandemic in Korean hospitals. However, an increasing trend in antibiotic use was observed for patients with mild-to-moderate COVID-19.

## Figures and Tables

**Figure 1 antibiotics-12-00198-f001:**
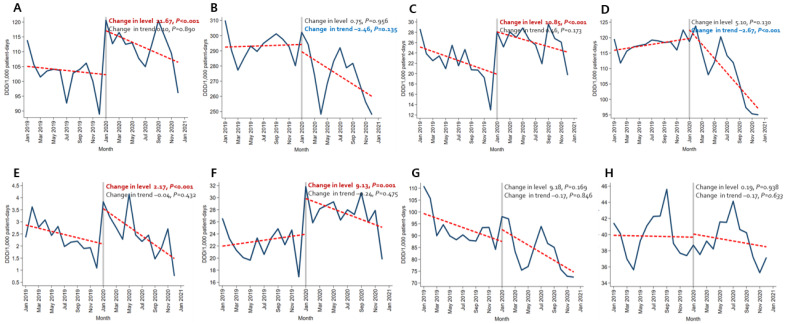
Changing patterns of antibiotic usage in COVID-19 patient management hospitals (the total number of COVID-19 patients in the bottom 10% of hospitals) during the COVID-19 pandemic. (**A**) Broad-spectrum antibiotics predominantly used for hospital-onset cases. (**B**) Broad-spectrum antibiotics predominantly used for community-acquired cases. (**C**) Antibacterial agents predominantly used for resistant gram-positive bacterial cases. (**D**) Narrow-spectrum beta-lactam agents. (**E**) Antibacterial agents predominantly used for extensive antibiotic-resistant gram-negative bacterial cases (**F**) Carbapenem. (**G**) Fluoroquinolone. (**H**) Metronidazole.

**Figure 2 antibiotics-12-00198-f002:**
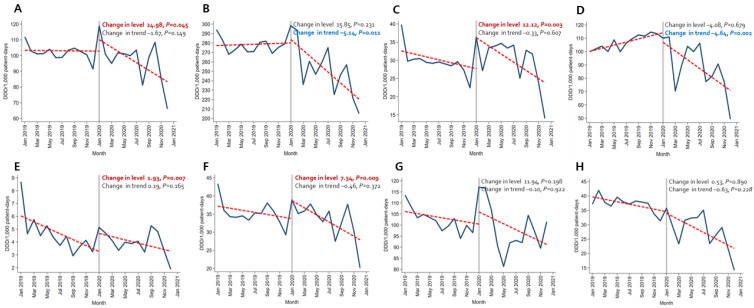
Changing patterns of antibiotic usage in the COVID-19 patient management hospitals (the total number of COVID-19 patients in the top 10% of hospitals) during the COVID-19 pandemic. (**A**) Broad-spectrum antibiotics predominantly used for hospital-onset cases. (**B**) Broad-spectrum antibiotics predominantly used for community-acquired cases. (**C**) Antibacterial agents predominantly used for resistant gram-positive bacterial cases. (**D**) Narrow-spectrum beta-lactam agents. (**E**) Antibacterial agents predominantly used for extensive antibiotic-resistant gram-negative bacterial cases. (**F**) Carbapenem. (**G**) Fluoroquinolone. (**H**) Metronidazole.

**Figure 3 antibiotics-12-00198-f003:**
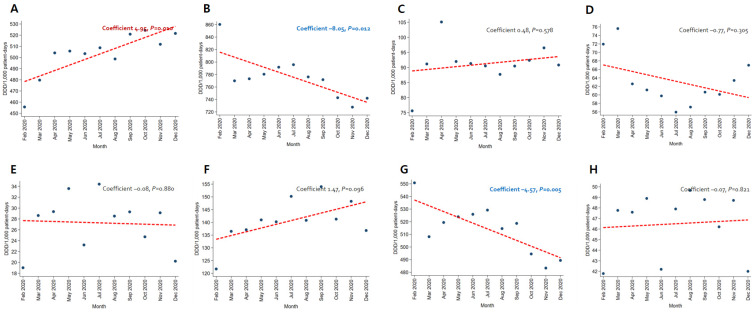
Changing patterns of antibiotic usage in patients with pneumonia during the COVID-19 pandemic. (**A**) Broad-spectrum antibiotics predominantly used for hospital-onset cases. (**B**) Broad-spectrum antibiotics predominantly used for community-acquired cases. (**C**) Antibacterial agents predominantly used for resistant gram-positive bacterial cases. (**D**) Narrow-spectrum beta-lactam agents. (**E**) Antibacterial agents predominantly used for extensive antibiotic-resistant gram-negative bacterial cases. (**F**) Carbapenem. (**G**) Fluoroquinolone. (**H**) Metronidazole.

**Figure 4 antibiotics-12-00198-f004:**
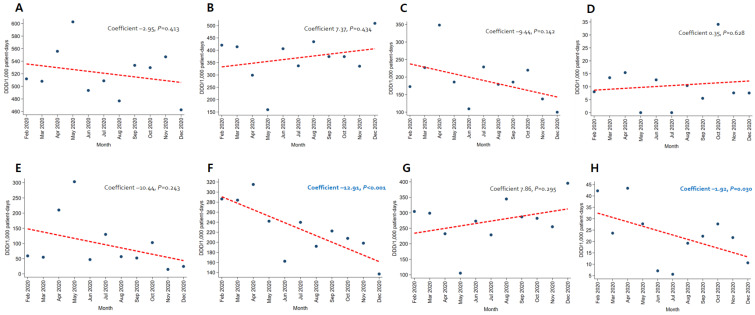
Changing patterns of antibiotic usage in patients with severe COVID-19 cases during the COVID-19 pandemic. (**A**) Broad-spectrum antibiotics predominantly used for hospital-onset cases. (**B**) Broad-spectrum antibiotics predominantly used for community-acquired cases. (**C**) Antibacterial agents predominantly used for resistant gram-positive bacterial cases. (**D**) Narrow-spectrum beta-lactam agents. (**E**) Antibacterial agents predominantly used for extensive antibiotic-resistant gram-negative bacterial cases. (**F**) Carbapenem. (**G**) Fluoroquinolone. (**H**) Metronidazole.

**Figure 5 antibiotics-12-00198-f005:**
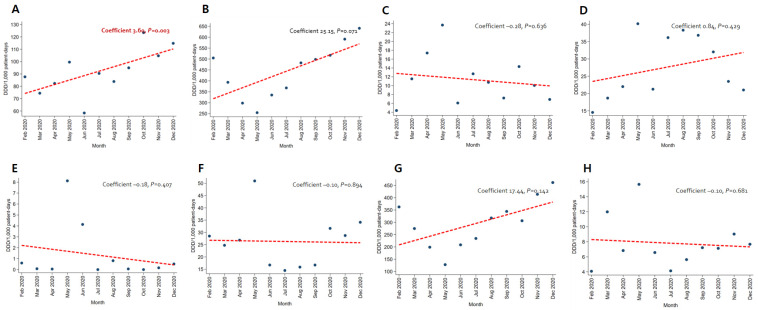
Changing patterns of antibiotic usage for patients with mild-to-moderate COVID-19 cases during the COVID-19 pandemic. (**A**) Broad-spectrum antibiotics predominantly used for hospital-onset cases. (**B**) Broad-spectrum antibiotics predominantly used for community-acquired cases. (**C**) Antibacterial agents predominantly used for resistant gram-positive bacterial cases. (**D**) Narrow-spectrum beta-lactam agents. (**E**) Antibacterial agents predominantly used for extensive antibiotic-resistant gram-negative bacterial cases. (**F**) Carbapenem. (**G**) Fluoroquinolone. (**H**) Metronidazole.

**Table 1 antibiotics-12-00198-t001:** Changing patterns of antibiotic usage in hospitals during the COVID-19 pandemic.

	Pre-COVID	Post-COVID	Change in Level	SE	95% CI	*p*	Change in Trend	SE	95% CI	*p*
Total antibacterial agents	697.82	652.87	12.57	25.74	−41.12 to 66.25	0.631	−9.95	3.24	−16.70 to −3.19	0.006
Broad-spectrum antibiotics predominantly used for hospital-onset cases	114.58	116.76	16.69	7.00	2.08 to 31.29	0.027	−0.86	0.93	−2.79 to 1.07	0.365
Broad-spectrum antibiotics predominantly used for community-acquired cases	295.23	278.58	2.06	11.70	−22.35 to 26.47	0.862	−3.89	1.34	−6.69 to −1.09	0.009
Antibacterial agents predominantly used for resistant gram-positive bacterial cases	31.41	31.73	8.60	2.53	3.33 to 13.88	0.003	0.05	0.38	−0.75 to 0.85	0.898
Narrow-spectrum beta-lactam agents	127.57	116.39	−5.36	3.58	−12.82 to 2.10	0.150	−2.42	0.42	−3.30 to −1.54	<0.001
Antibacterial agents predominantly used for extensive antibiotic-resistant gram-negative bacterial cases	5.48	5.49	1.99	0.71	0.51 to 3.47	0.011	−0.05	0.09	−0.23 to 0.14	0.588
Carbapenem	33.31	34.48	6.21	2.80	0.37 to 12.05	0.038	−0.10	0.40	−0.93 to 0.73	0.810
Fluoroquinolone	120.19	113.78	9.20	7.07	−5.55 to 23.96	0.208	−1.26	0.78	−2.89 to 0.38	0.124
Metronidazole	42.97	41.19	2.39	1.86	−1.48 to 6.27	0.213	−0.09	0.30	−0.71 to 0.54	0.775

Abbreviations: COVID = coronavirus disease 2019; SE = standard error; CI = confidence interval.

**Table 2 antibiotics-12-00198-t002:** Changing patterns of antibiotic usage in patients with pneumonia or COVID-19 during the COVID-19 pandemic.

	Pneumonia				COVID-19, Severe				COVID-19, Mild to Moderate			
	Coefficient	SE	95% CI	*p*	Coefficient	SE	95% CI	*p*	Coefficient	SE	95% CI	*p*
Total antibacterial agents	−9.73	1.72	−13.62 to −5.84	<0.001	−34.62	12.56	−63.03 to −6.21	0.022	25.55	10.88	0.93 to 50.17	0.043
Broad-spectrum antibiotics predominantly used for hospital-onset cases	4.95	1.51	1.52 to 8.37	0.010	−2.95	3.433	−10.71 to 4.82	0.413	3.60	0.91	1.55 to 5.66	0.003
Broad-spectrum antibiotics predominantly used for community-acquired cases	−8.05	2.56	−13.84 to−2.25	0.012	7.37	9.00	−13.00 to 27.74	0.434	25.15	12.29	−2.65 to 52.94	0.071
Antibacterial agents predominantly used for resistant gram-positive bacterial cases	0.48	0.83	−1.39 to 2.35	0.578	−9.44	5.87	−22.72 to 3.83	0.142	−0.28	0.58	−1.58 to 1.02	0.636
Narrow-spectrum beta-lactam agents	−0.77	0.71	−2.37 to 0.83	0.305	0.35	0.71	−1.24 to 1.95	0.628	0.84	1.01	−1.45 to 3.13	0.429
Antibacterial agents predominantly used for extensive antibiotic-resistant gram-negative bacterial cases	−0.08	0.54	−1.29 to 1.13	0.880	−10.44	8.36	−29.34 to 8.47	0.243	−0.18	0.21	−0.64 to 0.29	0.407
Carbapenem	1.47	0.79	−0.32 to 3.26	0.096	−12.91	2.42	−18.40 to −7.43	<0.001	−0.10	0.73	−1.75 to 1.55	0.894
Fluoroquinolone	−4.57	1.26	−7.42 to −1.72	0.005	7.86	7.08	−8.15 to 23.88	0.295	17.44	10.85	−7.11 to 41.98	0.142
Metronidazole	0.07	0.31	−0.63 to 0.77	0.821	−1.92	0.75	−3.61 to −0.23	0.030	−0.10	0.23	−0.62 to 0.42	0.681

Abbreviations: SE = standard error; CI = confidence interval.

## Data Availability

Further data are available upon reasonable request from the corresponding authors.

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
