# Peer review of "Analysis of Changes in Antibiotic Use Patterns in Korean Hospitals during the COVID-19 Pandemic"

_antibiotics, 2023, doi:10.3390/antibiotics12020198_

Round 1
Reviewer 1 Report
I would like to congratulate the authors for a comprehensive manuscript of this important global issue.
The following are my comments:
- The definition of pneumonia is very loose. Is it bacterial, viral or pneumonia secondary to COVID-19? How are they distinguished from each other? This can greatly confound the results of antibiotic usage.
- The categorization of antibiotics is also confusing. Levofloxacin can be used for both hospital and community acquired infections. Meropenem can be used as treatment for hospital acquired infection and in infections sec to extensively resistant GNR.
- It is also important to point out the time to diagnosis of COVID-19 and initiation of antibiotics. Some patients may develop bacterial pneumonia days to weeks after diagnosis of COVID. But I understand many patient data are not available for review.
- Would move Materials and Methods after the Introduction
Author Response
- The definition of pneumonia is very loose. Is it bacterial, viral or pneumonia secondary to COVID-19? How are they distinguished from each other? This can greatly confound the results of antibiotic usage.
à In fact, we included the codes relevant to viral pneumonia (J10, J11, and J12), because empirical antibiotics are likely to be applied to any type of pneumonia before causative pathogens are revealed. We added the above sentence in the method section in this round (Page 2).
As for the possibility of inclusion of post-COVID pneumonia, we added the below sentence in the limitation section.
(Page 12) Also, some case of pneumonia could be secondary complication of COVID-19 and might have been included in both COVID-19 and pneumonia group.
- The categorization of antibiotics is also confusing. Levofloxacin can be used for both hospital and community acquired infections. Meropenem can be used as treatment for hospital acquired infection and in infections sec to extensively resistant GNR.
à We understand your concern and agree with your comments. Because of it, In fact, experts in my country, including some of the authors in this article, discussed a lot over the 'antibiotic classification' issue. Throughout the Delphi method, the classification that was used for this study was adopted as the classification of national surveillance system for antimicrobial use (KONAS system). Further information about the classification can be obtained in the following article and we cited it in the manuscript.
Ref: Kim, B.; Yoon, Y.K.; Kim, D.-S.; Jeong, S.J.; Ahn, S.V.; Park, S.H.; Kwon, K.T.; Kim, H.B.; Park, Y.S.; Kim, S.-W., et al. Development of Antibiotic Classification for Measuring Antibiotic Usage in Korean Hospitals Using a Modified Delphi Method. Journal of Korean Medical Science 2020, 35, doi:10.3346/jkms.2020.35.e241.
- It is also important to point out the time to diagnosis of COVID-19 and initiation of antibiotics. Some patients may develop bacterial pneumonia days to weeks after diagnosis of COVID. But I understand many patient data are not available for review.
à We agree with your comment and we revised a limitation as follows:
(Page 12) Finally, the information on duration or combined antibiotic therapy was unavailable due to the nature of National Health Insurance claims data. This point also prevents under-standing of the timing of the initiation of antibiotics in COVID-19. A more detailed analy-sis of antibiotic use is warranted.
- Would move Materials and Methods after the Introduction
à We moved ‘Materials and Methods’ after the ‘Introduction’ section.
Reviewer 2 Report
The manuscript of Kim and colleagues investigated the changing patterns in antibiotic use in Korean hospitals during the COVID-19 pandemic with the data of National Health Insurance claims of inpatients from all hospitals in Korea between January 2019 to December 2020. They found a decreasing trend in overall antibiotic use during the COVID-19 pandemic and an increasing trend in antibiotic use in patients with mild to moderate COVID-19 in Korean hospitals. The results and conclusions are straightforward; however, the presentation of the data needs to be improved and the manuscript seems lack of impact. And some of the data analysis could be improved.
Major comments:
1. The introduction is not enough for the current knowledge of antibiotic use affected by COVID-19.
2. Table1A: Changing patterns of antibiotic usage in hospitals during the COVID-19 pandemic already showed the trends of described. Figures 1, 2 and tables 1B, 1C show top and bottom 10% hospitals data seems to be redundant.
3. Is it appropriate to use these data for coefficient calculations and determine the trend? There are some inconsistencies in this analysis (figure 3-5). For example, figure 3B has a negative coefficient, but figure 4B and figure 5B both have a positive coefficient. Figure 3 showed the data of all patients but figure 4 and figure 5 showed severe COVID-19 infection and mild to moderate COVID-19 infection, respectively. Theoretically, these data should be consistent? If not, the authors should discuss this in the manuscript.
4. The data compare to the same month of last year should be showed to see if the antibiotic use is month dependent.
Author Response
- The introduction is not enough for the current knowledge of antibiotic use affected by COVID-19.
à Thank you for your helpful comment. We added following sentences in the introduction section in this round.
(Page 2) Furthermore, a surge in COVID-19 cases which could have similar clinical features to bacterial pneumonia might influence the antibiotic usage pattern. For instance, when physicians face critically ill COVID-19 patients and the diagnosis of bacterial superinfection is uncertain, some of them might make a decision to use broad-spectrum antibiotics immediately rather than wait for a while.
- Table1A: Changing patterns of antibiotic usage in hospitals during the COVID-19 pandemic already showed the trends of described. Figures 1, 2 and tables 1B, 1C show top and bottom 10% hospitals data seems to be redundant.
à We removed Table 1B, 1C from the manuscript in this round.
- Is it appropriate to use these data for coefficient calculations and determine the trend? There are some inconsistencies in this analysis (figure 3-5). For example, figure 3B has a negative coefficient, but figure 4B and figure 5B both have a positive coefficient. Figure 3 showed the data of all patients but figure 4 and figure 5 showed severe COVID-19 infection and mild to moderate COVID-19 infection, respectively. Theoretically, these data should be consistent? If not, the authors should discuss this in the manuscript.
à There is a misunderstanding. Figure 3 showed patients with ‘pneumonia’ and Figure 4, 5 showed patients with ‘COVID-19’. We intended to compare the changing patterns of antibiotic use among pneumonia, COVID-19 (mild to moderate), and COVID-19 (severe); the changing pattern of antibiotic usage of pneumonia was similar to COVID-19 (mild to moderate).
- The data compare to the same month of last year should be showed to see if the antibiotic use is month dependent.
à We understand your concern. In fact, the first part of the results (Table 1, Figure 2, 3) included data not only from 2020 but also from 2019 in order to show the impact of COVID-19 pandemic on the overall antibiotic usage patterns in hospitals. In comparison, the second part of the results (Table 2, Figure 3-5) could not include data from 2019 because the COVID-19 pandemic began on Jan 2020 in Korea.
Round 2
Reviewer 1 Report
Thank you for your revision.
Reviewer 2 Report
The authors improved the manuscript and solved my concerns.